# Bone Regeneration Using Mesenchymal Stromal Cells and Biocompatible Scaffolds: A Concise Review of the Current Clinical Trials

**DOI:** 10.3390/gels9050389

**Published:** 2023-05-08

**Authors:** Federica Re, Elisa Borsani, Rita Rezzani, Luciana Sartore, Domenico Russo

**Affiliations:** 1Unit of Blood Diseases and Cell Therapies, Department of Clinical and Experimental Sciences, University of Brescia, “ASST-Spedali Civili” Hospital of Brescia, 25123 Brescia, Italy; domenico.russo@unibs.it; 2Centro di Ricerca Emato-Oncologica AIL (CREA), ASST Spedali Civili, 25123 Brescia, Italy; 3Division of Anatomy and Physiopathology, Department of Clinical and Experimental Sciences, University of Brescia, 25123 Brescia, Italy; elisa.borsani@unibs.it (E.B.); rita.rezzani@unibs.it (R.R.); 4Interdepartmental University Center of Research “Adaption and Regeneration of Tissues and Organs (ARTO)”, University of Brescia, 25123 Brescia, Italy; 5Department of Mechanical and Industrial Engineering, Materials Science and Technology Laboratory, University of Brescia, 25123 Brescia, Italy; luciana.sartore@unibs.it

**Keywords:** clinical trials, mesenchymal stromal cells, scaffolds, hydrogels, bone regeneration, tissue regeneration, human platelet lysate

## Abstract

Bone regenerative medicine is a clinical approach combining live osteoblast progenitors, such as mesenchymal stromal cells (MSCs), with a biocompatible scaffold that can integrate into host bone tissue and restore its structural integrity. Over the last few years, many tissue engineering strategies have been developed and thoroughly investigated; however, limited approaches have been translated to clinical application. Consequently, the development and clinical validation of regenerative approaches remain a centerpiece of investigational efforts towards the clinical translation of advanced bioengineered scaffolds. The aim of this review was to identify the latest clinical trials related to the use of scaffolds with or without MSCs to regenerate bone defects. A revision of the literature was performed in PubMed, Embase, and Clinicaltrials.gov from 2018 up to 2023. Nine clinical trials were analyzed according to the inclusion criteria: six presented in the literature and three reported in Clinicaltrials.gov. Data were extracted covering background trial information. Six of the clinical trials added cells to scaffolds, while three used scaffolds alone. The majority of scaffolds were composed of calcium phosphate ceramic alone, such as β-tricalcium phosphate (TCP) (two clinical trials), biphasic calcium phosphate bioceramic granules (three clinical trials), and anorganic bovine bone (two clinical trials), while bone marrow was the primary source of the MSCs (five clinical trials). The MSC expansion was performed in GMP facilities, using human platelet lysate (PL) as a supplement without osteogenic factors. Only one trial reported minor adverse events. Overall, these findings highlight the importance and efficacy of cell–scaffold constructs in regenerative medicine under different conditions. Despite the encouraging clinical results obtained, further studies are needed to assess their clinical efficacy in treating bone diseases to optimize their application.

## 1. Introduction

Musculoskeletal health problems due to osteoporosis, tumors, and fractures have been widely studied in recent years and bone is the most frequently transplanted tissue [1,2]. Each year an estimated 2.2 million individuals suffer fractures due to bone disease [3,4,5]. Bone tissue is capable of self-repair; however, it has been shown that the physiological processes are delayed or do not occur in some conditions. The current bone reconstruction procedures include autologous, allogeneic, and xenogeneic bone grafts [6]. These approaches have some disadvantages, including the volume of bone that can be harvested, and associated risks such as immunoreactions and infections. These factors increase treatment costs and patient discomfort [7,8]. Bone regenerative medicine has been introduced into clinical practice as an alternative therapeutic approach to overcome the obstacles related to the use of current bone graft substitutes, creating functional tissues instead of implanting non-living scaffolds [9]. In particular, bone tissue regeneration procedures are mainly based on culture-expanded autologous mesenchymal stromal cells (MSCs) associated with biomaterials that fulfill the requisites of osteogenesis (bone forming), osteoinduction (bone inducing), and osteoconduction (bone supporting) [10]. Several biomaterials have been used in combination with MSCs aiming to promote their adhesion, proliferation, and osteoblastic differentiation, as well as production of the collagen matrix that subsequently undergoes mineralization [11]. Moreover, biomaterials must be resorbable and allow the ingrowth of newly formed blood vessels from the neighboring tissues [12]. The majority of scaffolds investigated for bone regeneration applications are natural polymers (e.g., chitosan, fibrin, hyaluronic acid, and collagen) or synthetic polymers (e.g., polylactic acid, polycaprolactone); bioactive ceramics such as coralline, hydroxyapatite, tricalcium phosphate, sulphate, bioactive glass, and calcium silicate; and hybrid combinations of two or more materials with different properties in the form of co-polymers, polymer-polymer blends, or polymer–ceramic composites [13]. In this context, since synthetic and natural scaffolds have been described for bone tissue regeneration, this review explored whether they could be considered for clinical translation. Moreover, the development of scaffolds based on hydrogels is emerging since they offer the possibility of generating well-defined 3D fabricated tissue analogs to the native extracellular environment. Moreover, hydrogels can practically be cast into any shape, size, or form, typically under cytocompatible conditions, and contain a relatively low amount of dry mass (1–20%), causing little inflammation and foreign body reaction during degradation. These attributes make them potential candidates for bone regenerative medicine [14].

Usually, a bone tissue engineering approach consists of harvesting bone marrow from the iliac crest, abdominal fat, or other tissues, expanding those cells in vitro to a sufficient number, and then seeding the cells onto a suitable scaffold before implantation into the same patient for differentiation and/or tissue regeneration [15]. This approach allows the creation of an environment that drives and stimulates the cells to form new functional tissue that subsequently integrates into the existing tissue at the defect site [16,17]. The ability of MSCs to differentiate into various mesodermal cells, such as bone and cartilage lineages, and their immunomodulatory and anti-inflammatory capacities are widely acknowledged [18,19]. The term ‘mesenchymal stem cell’ is controversially discussed and some investigators in the field use the term ‘mesenchymal stromal cell’ instead of ‘mesenchymal stem cell’ since they are connective tissue cells that form the supportive structure of an organ. However, the precise definition of these cells remains a matter of debate and most scientists refer to them simply as “MSCs.” MSCs can be derived from different sources, such as bone marrow, adipose tissue, and oral tissues (dental pulp, periodontal ligament, gingiva) [20,21,22,23,24,25,26]. The therapeutic potential of these cells is rather derived from the release of growth factors, cytokines, and extracellular vesicles to their surrounding cells, which may favor bone regeneration and osteogenesis [27]. In the field of skeletal diseases, several preliminary clinical studies have demonstrated that MSCs are the most promising cell population [19]. To consider these cells in different clinical situations, MSCs must fulfill the definition criteria of advanced therapy medicinal products (ATMPs), as requested by the International Society for Cellular Therapy (ISCT) [15]. Many studies have used a mixture of different cell types, such as the stromal vascular fraction, which is not included among ATMPs [17]. Other groups have used a bone marrow aspirate concentrated by centrifugation in order to increase the number of mononuclear cells and consequently MSCs injected into the injury site [7]. These strategies, compared to ATMPs, may not provide a uniform cell product with exact proportions of defined subpopulations [28,29,30]. For MSC expansion, platelet lysate (PL) is a valid substitute for fetal bovine serum (FBS), avoiding the intrinsic risks of potential immune responses to animal antigens [31]. PL allows large-scale expansion of MSCs for clinical use, satisfying all the criteria of the ISCT [32]. Since the use of PL in regenerative medicine is emerging, standardization and quality control of platelet-derived products are needed for ex vivo expansion of ATMPs for transplantation, in compliance with good manufacturing practice (GMP) [33]. Process methodologies that should be standardized include the use of expired or fresh platelet concentrates, time and temperature of freezing, storage conditions, thawing temperature, the number of pooled units, centrifugation steps, filtration conditions, and determination of expiry date [34]. Additionally, donor characteristics such as age, sex, and platelet count influence the growth factor content in PL as well as the capacity to support MSC expansion [34]. Proteomic analysis by mass spectroscopy has been shown to be instrumental in the characterization and standardization of PL [4]. To prove their safety and efficacy, the key challenges are both the characterization and standardization of cell-based products and the determination of optimum patient characteristics [32]. Cell-based medicine may be produced by research physicians in hospitals as well as by pharmaceutical companies as for other novel biological medicines. This possibility for manufacture outside the standard medical paradigm makes cell-based ATMPs a ground for extensive study in order to clear the biology supporting their potential mechanisms of action. Today, the bone regeneration process can be implemented in some critical clinical situations or pathologies in which replacing the bone in a short time could accelerate physiological processes in order to improve the quality of life of patients [13]. However, in the future, the use of such approaches could be considered for all types of bone defects in several clinical fields. For this reason, it is predictable that the safety and efficacy of cell-based therapies are of major concern to regulatory authorities [21]. Against a documented lack of defined cellular populations used in clinical trials, this review aimed to examine the clinical trials published in the scientific literature between 2018 and 2023 that used scaffolds with or without MSCs for bone regenerative medicine in order to point out new trends in this field, which has reached the stage of patient application.

## 2. Materials and Methods

### 2.1. Data Source

This review was performed on clinical trials using scaffolds with or without MSCs for bone regenerative therapy. The search was performed on the electronic databases Pubmed, Embase, and ClinicalTrials.gov from 2018 to 2023 using the terms “(mesenchymal stromal cells) AND (bone tissue OR bone tissue regeneration) AND (hydrogel OR biomaterial OR scaffold)”.

### 2.2. Study Selection Process

Two independent reviewers conducted the screening process and analyzed the papers. First, the reviewers screened the resulting records by title and abstract, then the full text of selected manuscripts was screened entirely. The exclusion criteria were review articles or unregistered clinical studies. The inclusion criteria were clinical trials with randomized (RCT) or a non-randomized (CT) controlled trial designs. From the included studies, relevant data were extracted, summarized, and analyzed according to the purpose of the present work. In particular, the following data were evaluated: country, clinical phase, condition, follow-up, control used, MSC tissue source and characterization, and scaffolds for bone regeneration. Finally, the obtained results were summarized and compared.

## 3. Results

According to the search strategy, five clinical trials were found in PubMed, one in Embase, and three in ClinicalTrials.gov. All of these studies are summarized in the following paragraphs.

### 3.1. Overview of Published Clinical Trials with MSCs and Scaffolds (2018–2023)

A total of nine clinical trials were included in the analysis: six were present in the literature and three were reported on Clinicaltrials.gov (Table 1 and Table 2, Figure 1). Background information from each trial was summarized including clinical phase, condition, controls used, and follow-up. Studies were conducted in Spain [35], Mexico [36], Norway [37], the Czech Republic [38], Italy [39,40,41], Western Australia (NCT01742260), and Spain and Portugal [42]. One trial was reported in three publications [39,40,41]. Two clinical trials were at Phase I [35] (NCT01742260), two were at Phase I/II [37,39], one was at Phase IIa [38], and phase was not reported in one clinical trial [36,42]. The studies were prospective, open, non-randomized [35,37,39,40,41], randomized [36,42], interventional (NCT01742260), and in particular, one trial was multicentric [39,40,41]. The most common indications concerned the treatment of lumbar intervertebral degenerative disc disease (DDD) [35], the alveolar ridge [37,42], large skeletal defects [38], and fracture of long bones [29,30,31,32,33,34,35,36,37,38,39,40]. Clinical evaluation showed that the patients achieved lumbar fusion in up to five years using TCP and autologous MSCs [35]. An increase in bone mineralization in association with a decrease in inflammation were obtained thanks to the combination of MSCs from dental pulp and a collagen sponge scaffold in periodontal disease at the 6-month follow-up [36]. In particular, Hernandez et al. evaluated 10 controls for which only collagen scaffold without DPMSCs had been placed, observing a less impressive clinical outcome with respect to the cell-added scaffold group [36]. Successful ridge augmentation without adverse events in maxillofacial bone defects was pursued using BCP and autologous MSCs [37]. No significant differences were obtained using cancellous allografts compared to the combination of TCP and MSCs in promoting the healing of bone defects, whereas significant differences were documented following the implantation of TCP only and cancellous allografts in femoral bone defects [38]. Clinical and radiological evaluation confirmed complete bone consolidation in long bone non-unions at 12 months using biphasic calcium phosphate bioceramic granules and autologous MSCs [39,40,41]. In addition, some clinical trials reported no findings, results, or publications in ClinicalTrials.gov although the study’s expected completion dates were 2017 for NCT01742260, 2018 for NCT03682315, and 2022 for NCT03797963. Hydrogels are the new generation of scaffolds for bone reconstruction and DEXGEL Bone, a hydrogel used for alveolar ridge preservation, was shown to stimulate natural bone regeneration without side effects [42]. DEXGEL Bone is derived from the association of Bonelike by Biosckin^®^ (BL^®^), a glass-reinforced hydroxyapatite synthetic bone substitute, with a dextrin-based hydrogel named DEXGEL. In particular, Machado et al. compared the synthetic bone substitute BL^®^ (control) to its hydrogel-reinforced version, DEXGEL Bone (test), in the preservation of alveolar ridge dimensions following tooth extraction, demonstrating bone quantity and quality and primary stability of the implant [42]. Finally, considering all studies, the follow-up periods ranged between 1 and 60 months.

### 3.2. MSC Tissue Sources, Characterization, and Manipulation

Bone marrow cells represented the cells most commonly used, particularly autologous cells [35,37,38,39,40,41]. Cells from donors were also used in [36] and NCT01742260. The term ‘stem’ was much more commonly used than ‘stromal’. MSCs from bone marrow were expanded in vitro using GMP, according to common standard operating procedures (SOP), in a specific medium enriched with PL without animal products at different concentrations: 5% PL [35], and 8% PL [39,40,41]. Cells were used at two or three passages [35,38,39,40,41] and one passage [37]. Mononuclear cell isolation after density-gradient centrifugation was performed by Blanco et al. [35], while details of viability analysis by flow cytometry for the positivity of CD90, CD73, and CD105 markers and the negativity of CD14 and CD45 markers were reported by Gjerde et al. [37] and in the ORTHO-1 study [39,40,41]. Other tested markers were reported: MHCI, MHCII, CD16, CD45, CD34, CD19, CD3, CD14, and CD80 [38]. Additional analyses, such as bacteriological tests and cell attachment using the fluorescent dye DAPI on BCP, were also performed [39]. Gómez-Barrena et al. used crystal violet and a live/dead assay to indicate that MSCs were attached and alive [39,40,41]. Moreover, sterility, endotoxins, and mycoplasma were tested [39,40,41]. Additional quality controls were performed according to the requests of each country-specific national competent authority [39,40,41]. In addition, certificates of analysis included with the investigational medicinal product (IMP) were obtained by each national competent authority (NCA) of countries participating in the REBORNE consortium [39,40,41]. Cells were obtained from 40–100 mL [35], 15–20 mL [37], and 10–12 mL [42] of bone marrow aspirate while 50,000 white blood cells per cm^2^ of bone marrow aspirate in a culture chamber were used by researchers in the REBORNE consortium (ORTHO-1 study) and distributed to other units [39,40,41]. hDPSCs from young donors were only used in the research of Hernández-Monjaraz et al. [36]. The nature of the growth factors used in cell culture for cell expansion was not detailed, but it was declared that the experiments were conducted under the strict criteria of GMP, using animal-origin-free reagents [36].

### 3.3. Scaffolds for Bone Regeneration

The majority of scaffolds were composed of calcium phosphate ceramic, such as β-tricalcium phosphate (TCP) [35,38], biphasic calcium phosphate bioceramic granules [37,39,40,41], anorganic bovine bone (NCT03682315, NCT03797963), and hydrogel in association with hydroxyapatite [42]. A total of 1.5 × 10^−6^ cells/kg from the patient were mixed with 20 mL of TCP support [35], 20 × 10^6^ cells/cm^3^ were cultivated in BCP [37], 15 ± 4.5 × 10^6^ cells were applied onto an absorbable porous β-tricalcium phosphate sponge [38], and 20 × 10^6^ cells per mL were suspended in 10 mL solution with bioceramic granules to obtain the ORTHO-1 MSC tissue-engineered product [39,40,41]. Processing of bone biopsies, after scaffolds were seeded with implanted cells, was performed for histological staining using hematoxylin/eosin and Masson trichrome [39]. In addition, immunohistology was performed to identify macrophages with human CD68 primary antibody by Gómez-Barrena et al. [39,40,41]. Only in the research of Hernández-Monjaraz et al., 5 × 10^6^ DPMSCs dripped suspended in PBS were seeded onto a scaffold of lyophilized polyvinylpyrrolidone sponge^®^ (clg-PVP) in 0.5 cm^2^ fragments, while the control group only received PBS without DPMSCs [36]. Finally, in both groups, collagen membranes (Biomed extend^®^, ZimVie, CA, USA) were placed on the flap. Moreover, in the clinical trial of Herrmann (NCT01742260), the researchers created a skull-like scaffold composed of medical-grade bioceramic granules of beta-tricalcium phosphate by ChronOS (Synthes GmbH, Oberdorf) and cells (concentration not reported) were placed between the specially molded plastic scaffolds (PLA such as 70:30 poly(L-lactide-co-D,L-lactide) [36].

Not all of the studies used scaffolds in association with cells. In particular, the study by Machado et al. demonstrated the ability of the hydrogel to stimulate newly formed bone and biological compatibility with the host tissues [42]. The authors used DEXGEL, an in situ gelling hydrogel with oxidized dextrin as the base, as a moldable carrier of BL^®^ granules in the management of alveolar bone regeneration. BL^®^ is a synthetic bone graft designed to mimic the inorganic composition of bone [42]. Even if no cells were used in association with DEXGEL Bone, BL^®^ (control) was mixed with autologous blood previously extracted from the alveolar defect and applied with a spatula [42]. Moreover, two other studies tested scaffolds without cells for sinus floor augmentation but no results were reported. The first study used xenograft bovine hydroxyapatite (BioOss) (NCT03682315) with contralateral active control of the biphasic phycogenic biomaterial and autogenous cortical bone. The second study added BioOss to the autogenous cortical bone (NCT03797963) with contralateral active control of the porcine bone mineral (Symbios Xenograft) mixed with autogenous cortical bone.

## 4. Discussion

In the last year, tissue engineering approaches have garnered great interest for bone regeneration [43]. A wide variety of biomaterials for bone regeneration has been used preclinically to provide osteogenesis, osteoinduction, and osteoconduction, although they have been rarely tested in clinical trials [44,45]. Biomaterials approved for clinical use are better candidates for clinical trials in combination with ATMPs, such as expanded MSCs cultured in GMP [43]. A mandatory step required by regulatory agencies is the preclinical evaluation of the association of the selected biomaterial and the cell product before starting the clinical trial. MSCs are reported to be the most frequently studied stem cells in clinical trials, even if not all of the clinical trials used cells in combination with support. A recent analysis examined >1000 stem cell clinical trials, of which 50% were early phase investigations (phases I–II) [46]. The main applications of bone tissue engineering approaches are non- and mal-unions of bone, osteonecrosis/avascular necrosis (e.g., femoral head, humerus, talus, and knee), orthopedic surgery (e.g., trauma applications, spinal fusion, fracture repair, and revision of endoprostheses), and dentistry applications (e.g., dental implants, cranial and maxillofacial applications) [47].

As the advances in bone tissue engineering move toward application in the clinical setting, the demonstration of the therapeutic efficacy of these novel scaffold designs is critical.

This review aimed to analyze the current clinical trials in the last five years that explored the development of appropriate protocols for bone tissue engineering to produce clinical-grade biomedical devices based on scaffolds with or without MSCs. The efficacy of a tissue engineering approach, based on biomaterial with or without MSC transplantation, was confirmed in all clinical trials. Particularly, the majority of the reported clinical trials employed properly ISCT-defined MSCs produced in GMP facilities. The SOP included a safety screening for the end product, including cell content, immunophenotype, sterility, endotoxins, and karyotype. Generally, different cell doses have not been tested but are adapted from preclinical findings. In fact, the dose of MSCs provided to a site may be a critical variable in the success of therapy. The number of cells seeded was a strategic decision aiming to minimize the risk of sub-therapeutic effects [39,40,41]. Most of the clinical trials discussed here used MSCs derived from bone marrow, except for the study by Hernández-Monjaraz et al., which employed hDPSCs from young donors [36]. The source of the cells also affects their biological properties and, in turn, may have an impact on bone regeneration. Bone marrow has been the primary source of MSCs for the majority of clinical applications since its aspiration is considered a safe procedure for obtaining cells compared to other anatomical locations [41,42]. hDPSCs should be an easy way to obtain MSCs, although they are used less often.

Four clinical trials used autologous cells, while two clinical trials used MSCs from donors. When studied for their therapeutic safety and efficacy, the impact of allogenic MSCs was somewhat comparable yet less efficacious than those from autologous sources. In addition, the healing effects (such as anti-inflammatory and bone regeneration) of allogenic MSCs are less than those of autologous MSCs [18].

The MSCs were expanded using no osteogenic factors, and no osteogenic factors were used in the clinical procedure. MSCs were cultured using platelet derivates such as PL [37]. It has been demonstrated that the administration of PL is beneficial as it induces the migration of MSCs directly to the site of injury or surgery [31]. Moreover, PL contains growth factors such as PDGF, TGF-beta, IGF, VEGF, and FGF, which are known to be involved in the regenerative processes of bone tissue [32]. However, PL varies between each person and due to the many variables involved in the manufacturing process [34]. Not all of the clinical trials reported the exact number of donors from which PL was derived [35], for this reason the PL product is still a subject of debate due to the many variables involved in its manufacturing process [33]. Even if several groups tried to standardize the PL manufacturing procedure, several open questions must be answered [34]. For this reason, there is a need to standardize PL in compliance with GMP so that each patient can obtain the full benefits from platelet derivatives [34].

Even if no osteogenic factors were added, a positive effect on osteogenic “pre-differentiation” of MSCs using PL as a supplement during the isolation and expansion phases cannot be excluded [34]. Notably, neither the pre-differentiation of MSCs in osteoblasts nor a mix of MSCs and pre-differentiated MSCs in osteoblasts are mandatory steps in these regenerative protocols. This consideration underlines the ability of MSCs to differentiate themselves and also to recruit different cell types in situ for tissue regeneration.

Only one clinical trial analyzed the biopsy specimens taken 4–6 months after augmentation showing new bone formation, with blood supply and without inflammatory cells [37]. Moreover, MSCs seem to have a positive effect on neighboring soft tissues, such as keratinized mucosa, and contribute to wound healing. Meanwhile, other authors sustained that a biopsy could not be performed as a standard procedure, particularly in uncomplaining patients [38]. TCP scaffolds were the most employed in association with MSCs, since this bone substitute is easy to use and presents appropriate physical and chemical characteristics similar to human bone [35,38]. TCP scaffolds are considered the “gold standard” of synthetic bone grafts and are generally resorbed in 13–20 weeks after implantation [47]. TCP scaffolds have been shown to induce bone regeneration and significant differences were documented compared to cancellous impaction allografting. However, no significant difference emerged in bone defect healing following the implantation of TCP for expanded autologous MSCs compared to cancellous impaction allografting [38].

TCP has been associated with hydroxyapatite (HA) for bone formation in vivo using a synthetic scaffold of biphasic calcium phosphate (BCP) to obtain the advantages of both materials. Mixing HA with TCP allows a range of resorption times and mechanical properties, ensuring that the material is stable and can promote bone growth [47]. Thanks to these properties, a TCP scaffold can be used in large bone defects and load-bearing areas [47]. TCP and MSCs were used as a new therapeutic approach to regenerate alveolar bone defects and for the treatment of non-unions in the tibia, femur, and humerus (ORTHO-1 multicentric clinical trial) [35,39,40,41]. From the considered studies, it was deducible that the nature of the scaffold (HA and TCP) is a priority with respect to aspects of the skeleton structure not considered in detail when using these materials, such as the porosity. The clinical trial of Herrmann (NCT01742260) also introduced the strategy of mimicking skull tissue organization, proposing medical-grade bioceramic granules of ß-tricalcium phosphate added with cells placed between plastic scaffolds (PLA). Unfortunately, no results have been posted. Finally, the use of a collagen scaffold was a possibility that was considered [25].

Even if most studies used TCP, the latest studies and clinical trials preferred using hydrogels for various medical applications [35]. This aspect demonstrated that tissue engineering strategies evolve, and although scaffolds may not be constituted by natural materials such as TCP or HA in the future, they could be based on natural or synthetic materials not typical of bone tissue. These novel types of scaffolds offer the possibility of customization and demonstrated recent progress in strategies for constructing biomimetic scaffolds.

Hydrogels can practically be cast into any shape, size, or form, and Machado et al. confirmed that changing the physicochemical properties of a bone substitute hydrogel can influence implant stability [42]. In this way, the current use of hydrogels adds the possibility of customizing the skeleton. In addition, the presence of an adequate pore dimension in the hydrogel favors osteointegration, osteoconduction, and degradation, allowing bone ingrowth in the interspaces, as it enables blood vessels and cell infiltration, exchange of proteins and nutrients, and waste clearance [35]. Hydrogels addressed to bone regeneration are generally based on extracellular matrix components (collagen, fibrin, polypeptides) that provide an adhesive surface for the cells, while polysaccharides such as agarose, alginate, hyaluronic acid, chitosan, and hydroxylpropylmethyl cellulose are interesting candidates for cell encapsulation [34]. Generally, the scaffolds used in these clinical trials are considered “second-generation biomaterials” for bone regeneration [47]. These scaffolds develop the concept of “bioactivity,” inducing a beneficial physiological response in the body, mainly including synthetic and naturally derived biodegradable polymers, calcium phosphates, calcium carbonate, calcium sulfates, and bioactive glasses. There is additionally an increasing emphasis on biodegradability, such that once a scaffold is resorbed, the cells seeded within the scaffold interact with the surrounding environment [48,49,50,51,52,53,54].

In order to define bone healing, robust criteria are needed. For example, E. Gómez-Barrena et al. [50] elaborated the REBORNE healing scale in tibia, humerus, and femur non-unions [51], in which computed tomography (CT) imaging is considered the gold standard to assess bone healing. Compared to the previous scoring system, named RUST (Radiographic Union Scale for Tibial fractures), the REBORNE score simplified bone healing into three stages, with advantages in reliability, repeatability, and validity that have been widely verified [52,53,54,55,56].

Three of the clinical trials had some limitations, such as the low number of patients included [35,37,42]. For this reason, the promising results should be interpreted with caution. However, the duration of follow-up was extended over five years by Blanco et al. [35]. This was also the final part of a research line that started with in vitro characterization and preclinical evaluation in rabbits [35]. In vivo biocompatibility and safety were demonstrated through the assessment of the inflammatory response in rat subcutaneous implants, subacute systemic toxicity, and skin sensitization using rodent models in a previous preclinical study by Machado et al. [42]. However, a large study cohort and longer follow-up period are necessary for application in a standard clinical setting. Generally, follow-up is carried out up to the twelfth month. Nevertheless, the follow-up periods were sufficient to reveal the evident healing of bone defects. In addition to the timing to establish consolidation, other factors were considered, including gender and tobacco use [39,40,41]. While gender did not influence clinical consolidation, higher consolidation scale values were seen in non-smoking patients at 6 and 12 months of follow-up [39,40,41].

A major future challenge for any proposed biomaterial combined with ATMPs is that it needs to be compared with other gold standards for bone healing augmentation, such as bone autograft. Moreover, since only one clinical trial was completed, further studies are needed to assess the clinical usefulness of bioengineered scaffolds with a cell product for treating bone diseases. In order to confirm the efficacy of cell/scaffold-based therapy, more clinical trials should be conducted. Moreover, the cost-effectiveness of the therapy should be also investigated. Taken together, these data suggest that regenerative medicine based on biomaterials, especially combined with bone marrow-derived mesenchymal stromal cells, will be a promising tool for the treatment of various diseases and the development of medicine in the future.

## 5. Conclusions

With the increasing age of the population, musculoskeletal health problems due to osteoporosis, tumors, and fractures are becoming more frequent. This review underlined overall positive findings about the transplantation of MSCs, as ATMPs, together with biocompatible scaffolds for a variety of bone problems. All clinical trials confirmed the effectiveness of a tissue engineering approach to treat bone diseases, including fractures of the femur, tibia, and humerus, but also DDD, deep infra-bony defects, and maxillofacial bone defects. The majority of the clinical studies used MSCs from bone marrow, reflecting the fact that they are the most accepted cell source for treating bone diseases due to the proliferation, differentiation, immunogenicity, and abilities of these cells. In combination with MSCs, bioactive materials have been demonstrated to significantly enhance bone formation, showing positive results on the whole. Most of the studies used TCP scaffolds; however, the future direction of regenerative medicine involves the use of hydrogels to control the shape, porosity, surface morphology, and size of the scaffold. Our study findings are in line with several recommendations proposing the advancement of cell therapy characterization and standardization for clinical transformation. We are confident that with the development of technologies and procedures, some of the barriers will be crossed, thereby promoting the clinical application of tissue engineering approaches.

## Figures and Tables

**Figure 1 gels-09-00389-f001:**
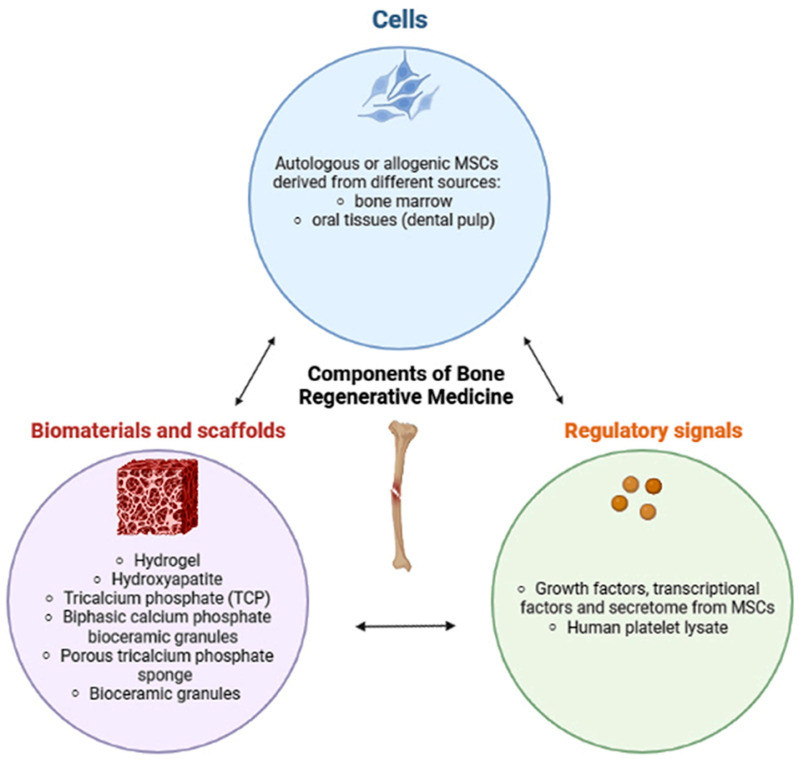
Bioengineering strategy beyond bone regeneration in clinical practice. The nature and structure of the scaffolds have great importance to support cell growth. The MSCs represent a source of growth factors, cytokines, and extracellular vesicles to their surrounding cells, which may favor bone regeneration and osteogenesis [10].

**Table 1 gels-09-00389-t001:** Summary of clinical trials using MSCs and scaffolds for bone regeneration from 2018 to 2023.

Cells	Scaffolds	Condition	Number of Patients (Age)	Number of Cells Seeded (Scaffolds Dimensions)	Follow Up	Control	Evaluation Methods and Outcomes	References
Autologous bone marrow-derived MSCs	ß-tricalcium phosphate (TCP)	Lumbar degenerative disc disease (DDD) at L4-L5 or L5-S1	11 (18–65)	1.5 × 10^−6^ cells/kg from the patient (20 mL of TCP)	1, 3, 6, 12, 60 months	_	Radiography and clinical evaluation revealed that 80% of patients achieved lumbar fusion in up to five years. Both the visual analog scale (VAS) and the Oswestry disability index (ODI) improved after surgery. The Short-Form Health Survey (SF-36) evaluated the physical and mental status that showed a significant improvement in the first year after surgery. There were no adverse effects related to cell implantation.	Blanco et al., 2019 [35]
MSCs obtained from the dental pulp of two male patients ages 7 and 8 and a 10-year-old patient (hDPSCs)	Scaffold of lyophilized collagen-polyvinylpyrrolidone sponge (Fibroquel; Aspid, Mexico City, Mexico)	Deep infra bony defect ≥ 4 mm deep caused by periodontal disease	22 (55–64)	5 × 10^6^ hDPSCs (0.5 cm^2^)	6 months	11 scaffolds without hDPSCs	Increase in the bone mineral density of the alveolar bone; increased salivary superoxide-dismutase and decreased levels of salivary IL1β	Beatriz Hernández-Monjaraz et al., 2020 [36]
Autologous bone marrow-derived MSCs	Biphasic calcium phosphate granules (BCP)	Maxillofacial bone defects	11 (52–79)	20 × 10^6^ cells (1 cm^3^)	1, 2, 4,12 months	_	All patients had successful ridge augmentation and an adequate amount of bone for dental implant installation without adverse events. The alveolar ridge increased both in width and volume.	Gjerde et al., 2018 [37]
Autologous bone marrow-derived MSCs	ß -tricalcium phosphate (TCP)	Femoral bone defect	37 (44–75)	15 ± 4.5 × 10^6^ cells (dimensions not reported)	6 weeks, 3, 6, 12 months	Group A: 19 patients with ß -TCP and autologous MSC, group B: 19 patients with ß -TCP alone, group C: 19 patients with cancellous allografts only	The combination between TCP and MSCs appears safe and promotes the healing of bone defects. No significant differences were observed between groups A and B. Significant differences were observed between group B and C. Adverse events emerged from the demanding and extensive character of revision hip replacement without a causal relationship to the suspension of autologous MSCs.	Pavel Sponer et al., 2018 [38]
Autologous bone marrow-derived MSCs	Biphasic calcium phosphate bioceramic granules (BCP)	Long bone non-unions (fractures of the femur, tibia, and humerus)	28 (3 months), 27 (6 months), 25 (12 months) (18–65). E. Gómez-Barrena et al., 2020 [25] 26 (18–65). E. Gómez-Barrena et al., 2020 [26] 28 (18–65). E. Gómez-Barrena et al., 2020 [27]	20 × 10^6^ cells (5–10 cc of bioceramic granules)	3, 6, 12 months For E. Gómez-Barrena et al., 2020 [27] subgroup analysis of gender, tobacco use, time since the original fracture	_	The ATMP combined with the bioceramic was surgically delivered to the non-unions, and 26/28 treated patients were found radiologically healed at one year (3 out of 4 cortices with bone bridging). E. Gómez-Barrena et al., 2020 [25] The REBORNE bone healing score, defined to perform an evaluation of long bone non-union consolidation in radiograph and computed tomography (CT), proved valid to assess consolidation against CT measurements with a concordance correlation of 79% and an accuracy based on ROC curves of 83%. E. Gómez-Barrena et al., 2020 [26] The clinical and radiological evaluation confirmed bone consolidation at 3 months (25%), 6 months (67.8%), and 12 months (92.8%), with lower consolidation scores in smokers. Femur, humerus, and tibia showed consolidation at one year. E. Gómez-Barrena et al., 2020 [27]	E. Gómez-Barrena et al., 2020 [39] E. Gómez-Barrena et al., 2020 [40] E. Gómez-Barrena et al., 2020 [41]
Bone marrow-derived MSCs from donor	Medical grade bioceramic granules of beta-tricalcium phosphate by ChronOS (Synthes GmbH, Oberdorf) placed between specially moulded plastic scaffolds (PLA such as 70:30 polyia (L-lactide-co-D,L-lactide) and insert the sandwich into the skull.	Cranial defect <80 mm diameter	10 (18–80)	Not reported	12 months	_	Quantitative bone density of the tissue-engineered construct and adjacent bone from CT scan at 12 months. Assessment of cosmesis by photography. NO outcomes	No publication, no results posted estimated study completion date 2017

**Table 2 gels-09-00389-t002:** Summary of clinical trials using scaffolds without cells for bone regeneration from 2018 to 2023.

Cells	Scaffolds	Condition	Number of Patients (Age)	Follow Up	Control	Evaluation Methods and Outcomes	References	Registration ID and Country
No cells	Anorganic bovine bone (BioOss Xenograft)	Bilateral Maxillary Sinus Floor Augmentation	8 (>18 years)	6 months	Active Comparator (contralateral: biphasic phycogenic biomaterial and autogenous cortical bone)	CBCT scans before the sinus floor elevation and 6 months later before implant placement to calculate vertical bone height change from the crestal bone to the floor of the maxillary sinus. Histomorphometric quantification of new mineralized tissue, non-mineralized tissue and remaing graft particles in a bone biopsy collected 6 months after the grafting procedure. NO outcomes.	No publication, no results posted. Actual study completion date 2018	NCT03682315 Responsable: Pablo Galindo-Moreno, Universidad de Granada, Spain Phase not applicable
No cells	Anorganic bovine bone (BioOss Xenograft) + autogenous cortical bone	Bilateral Maxillary Sinus Floor Augmentation	10 (>18 years)	6–12–18 months	Active Comparator (contralateral: Porcine bone mineral (Symbios Xenograft) + autogenous cortical bone)	CBCT scans after the sinus floor elevation and 6–12–18 months later before implant placement to calculate vertical bone height change. Histomorphometric quantification of new mineralized tissue, non-mineralized tissue and remaing graft particles in a bone biopsy collected 6–12–18 months after the grafting procedure. NO outcomes.	No publication, no results posted. Actual study completion date 2022	NCT03797963 Responsable: Pablo Galindo-Moreno, Universidad de Granada, Spain Phas not applicable
No cells but BL^®^ was mixed with autologous blood previously extracted from the alveolar defect and applied with a spatula.	DEXGEL Bone: Bonelike by Biosckin^®^ (BL^®^), a glass-reinforced hydroxyapatite synthetic bone substitute, in association to dextrin-based hydrogel, DEXGEL	Alveolar ridge preservation	12 (above 18 years)	6 months	BL^®^ granules (250–500 μm) were administered to 6 randomized participants whereas the other 6 received DEXGEL Bone.	Both treatments showed good osseointegration. DEXGEL Bone exhibited increased granule resorption accompanied by a tendency for more new bone ingrowth compared to the BL^®^ group. DEXGEL was rapidly resorbed and accelerated BL^®^ resorption as well, freeing up space that favored new bone ingrowth, without compromising mechanical support. The healing of defects was free of any local or systemic complications.	Machado et al., 2023 [42]	EUDAMED: CIV-PT-18–01–02,705. RNEC: 30122. Portugal Phase not reported

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
