# Peer review of "Bone Regeneration Using Mesenchymal Stromal Cells and Biocompatible Scaffolds: A Concise Review of the Current Clinical Trials"

_gels, 2023, doi:10.3390/gels9050389_

Round 1

Reviewer 1 Report

The reviewer can perceived that it is a paper review. The presentation of this paper review is good, in overall. However, the reviewer wish to suggest  the number of reference that reviewed by authors can be increased to 70 papers, if it doesn't burden the main author. There is value added for this paper review. 10 years ago, the paper review is commonly involve up to 100 papers to increase the credibility of paper review.

Author Response

REPLY to Reviewer 1

The reviewer can perceived that it is a paper review. The presentation of this paper review is good, in overall. However, the reviewer wish to suggest the number of reference that reviewed by authors can be increased to 70 papers, if it doesn’t burden the main author. There is value added for this paper review. 10 years ago, the paper review is commonly involve up to 100 papers to increase the credibility of paper review.

REPLY: We thank the Reviewer for the precious comments. We chose a short temporal window to find the very last news in this field. We agree with the suggestion of the Reviewer regarding the increase of the reference in order to add value to the Review.

The number of papers in the Reference Section has been improved with the most recent paper published in this field, nevertheless, the suitable papers are overall limited in number. So, we add  8 papers reaching 56 total references; the new papers have the following reference numbers: 1, 2, 3, 4, 5, 9, 17, 18. While references numbers 13, 14, 15, 16, 20 have been changed with more updated references.

Reviewer 2 Report

In this manuscript, the authors consider such an important and urgent problem of modern regenerative medicine as the use of mesenchymal stromal cells in clinical practice. The relevance of this study is beyond doubt. Adding the "Materials and Methods" section to the review is only an advantage. In the process of studying this manuscript, I had the following questions and comments:

1. This manuscript is under consideration in the journal "Gel", there are scaffolds in the title of the manuscript, but very little attention is paid to synthetic and natural scaffolds in the work itself. The structure and properties of skeletons for tissue engineering of bone tissue are no less important for bone tissue regeneration than cells. Moreover, the authors in Table 1 indicate the sources of publications on the use of wireframes without a wireframe in clinical practice.

2. The clinical use of materials and cells requires a serious approach and justification of the choice of this type of cells or a certain type of framework. I would like to see in the review information about when it is necessary to use such approaches to restore bone tissue.

3. It is not very clear what the authors wanted to demonstrate using the information presented in Figure 1. Particular attention is paid to regulatory factors, although there is very little information in this area in the review.

4. The whole article is mainly descriptive. This is quite acceptable, but the "Discussion" section should contain a discussion, and not a retelling of the information that was presented in the article above.

Author Response

REPLY to Reviewer 2

In this manuscript, the authors consider such an important and urgent problem of modern regenerative medicine as the use of mesenchymal stromal cells in clinical practice. The relevance of this study is beyond doubt. Adding the "Materials and Methods" section to the review is only an advantage.

In the process of studying this manuscript, I had the following questions and comments:

  1. This manuscript is under consideration in the journal "Gel", there are scaffolds in the title of the manuscript, but very little attention is paid to synthetic and natural scaffolds in the work itself. The structure and properties of skeletons for tissue engineering of bone tissue are no less important for bone tissue regeneration than cells.

REPLY: We thank the Reviewer for the appropriate suggestion. In this work, the use and nature of the scaffold have been considered important as well as the cell source because we think that these two elements work in synergy for an optimal result. In fact, the keywords used during the literature search include also, at the same time, these two elements.

Moreover, the nature of the scaffold has been described in Introduction (line 65-70) and in Discussion (line 409-412) in which the scaffolds have been preclinically tested. The following sentence has been added in the Discussion: “In the considered studies, it is deducible that the nature of the scaffold (HA and TCP) is a priority with respect to the skeleton structure not considered in detail using these materials, such as the porosity.” (line 363-365).

We modified the following sentence in the Introduction: “In this context, the development of scaffolds based on hydrogels….” TO “In this context, since synthetic and natural scaffolds have been described for bone tissue regeneration, this review explored if both could be considered for clinical translation. Moreover, the development of scaffolds based on hydrogels” (line 70-72).

We added the following sentence in the Discussion: “In this way, the current use of hydrogels adds the possibility to customize the skeleton.” (line 380-381).

Moreover, the authors in Table 1 indicate the sources of publications on the use of wireframes without a wireframe in clinical practice.

REPLY: We thank the Reviewer for the observation. Table 1 and Table 2 have been revised updating the column “Outcomes” as “Evaluation Methods and Outcomes” specifying when the results of some studies have not been posted or published adding “NO outcomes”. Table 1 and 2 with track changes are uploaded as separated word files, while in the text the images are already corrected as the final form without track changes. 

  1. The clinical use of materials and cells requires a serious approach and justification of the choice of this type of cells or a certain type of framework. I would like to see in the review information about when it is necessary to use such approaches to restore bone tissue.

REPLY: We thank the Reviewer for the appropriate observation. In the text, the information about the regenerative potential of MSC in bone reconstitution has been reported in Introduction (line 84-86). Moreover, the following sentence has been added to the Introduction: “Today the bone regeneration process could be implemented considering some critical clinical situations or pathologies in which the bone should be replaced in a short time accelerating the physiological processes in order to improve the quality of life in patients [13]. However, in the future, the use of such approaches should be considered for all types of bone defects in several clinical fields.” (line 124-128)

  1. It is not very clear what the authors wanted to demonstrate using the information presented in Figure 1. Particular attention is paid to regulatory factors, although there is very little information in this area in the review.

REPLY: The authors wanted to summarize the strategy beyond the use in clinical practice of appropriate scaffolding and cells. The legend of Figure 1 has been changed as follows: “Bioengineering strategy beyond bone regeneration in clinical practice. The nature and the structure of the scaffolds have great importance to support cell growth. The MSCs represent a source of growth factors, cytokines, and extracellular vesicles to their surrounding cells that may favor bone regeneration and osteogenesis [10].” (Figure 1)

  1. The whole article is mainly descriptive. This is quite acceptable, but the "Discussion" section should contain a discussion and not a retelling of the information that was presented in the article above.

REPLY: We thank the Reviewer for the observation. The discussion has been modified and improved as you can see in the track changing text. In particular, we added the paragraph discussing the following topics:

- autologous and allogenic cells used in clinical trials  (line 315-319)

- use of HPL (line 328-331)

- predifferentiation protocol of MSCs (line 338-341)

- the nature of scaffold (line 363-365)

- the novel types of scaffold (line 372-377 and 380-381)

Reviewer 3 Report

Authors have conducted this review on the current status of the assessments of MSCs and biocompatible scaffolds for bone and tissue regenerations. They have mentioned in both the abstract and the introduction that this is a systematic review. However, as mentioned below  section by section , I believe that this review has simply not been conducted in a “systematic” manner. 

Abstract:

Lines 27, 28, and 29: 

“Data were extracted covering background trial information including location, clinical phase, condition, follow-up, control used, MSCs tissue source and characterization, and scaffolds used.”: 

I find including your primary and secondary outcomes in the abstract unnecessary. These details belong to the “methods” and the “results” sections. Instead, I highly suggest authors detail their study questions according to the PICO format in their abstract so that authors and reviewers would have a proper comprehension of the reviewers’ main goals and concerns.

Lines 29, 30, 31, and 32:

“Overall, these findings highlight the importance and the efficacy of cell-scaffolds constructs in bone regenerative medicine in different conditions. The majority of studies used calcium phosphate-based scaffolds and hydrogels for tissue regeneration, while bone marrow was the primary source of cells.”

I strongly advice not using vague and undetailed sentences and reports in your abstract. If authors want to report any part of their results in the abstract, the proper way would be including actual numbers or percentages instead of descriptive vague statements. 

Keywords:

I suggest adding “hydrogels” to the list of keywords.

Introduction:

·      Lines 40 and 41: 

“Musculoskeletal health problems due to osteoporosis, tumors, and fractures are widely studied in recent years and bone is the most frequently transplanted tissue”. Please refer to a newly published study (preferably a review of literature) for this statement. 

·      Lines 41 and 42:

“Each year an estimated 2.2 million individuals suffer a fracture due to bone disease”. Since this statement directly indicated the epidemiologic state of the mentioned disease/accident, I highly suggest authors include more studies that are newer than reference “1”. Authors do not have remove reference “1”, but add newer references along with reference “1” for this epidemiologic statement. 

·      Lines 48, 49, 50 and 51:

“Bone regenerative medicine has been introduced into clinical practice as an alternative therapeutic approach to overcome the obstacles related to the use of current bone graft substitutes to create functional tissues instead of implanting non-living scaffolds” 

The reference assessed for these statements is dated and needs to be updated.

·      Lines 74, 75 and 76:

“This approach allows the creation of an environment that drives and stimulates the cells to form new, functional tissue that subsequently integrates into the existing tissue at the defect site” 

The reference assessed for these statements is dated and needs to be updated.

·      Lines 84, 85, 86, and 87:

“The therapeutic potential of these cells is rather derived from the release of growth factors, cytokines, and extracellular vesicles to their surrounding cells that may favor bone regeneration and osteogenesis”. 

The referred study for this statement is from 2012. When the therapeutic potential of MSCs is mentioned, I strongly believe that using a dated study does not do this subject any justice. Authors need to search for newly published studies on this subject and refer to them along with reference “11”.

·      Lines 87 and 88:

“In the field of skeletal diseases, several preliminary clinical studies have demonstrated that MSCs are the most promising cell population [18,19].” 

References number 18 and 19, are assessed before reference number 12. Authors need to re-adjust all of their references and their references list. I strongly advice utilizing referencing tools (e.g., Endnote) to prevent and correct these kinds of mistakes. 

Overall, authors need to refer to newly-published studies when the history, epidemiology, or previous works done of a particular subject are mentioned. Otherwise, their judgements and reports would not be justifiable. In addition to the mentioned references above, references number 13, 15, 16, and 20, all need to be updated to newer studies. I do understand that sometimes referring to old studies is justifiable when authors are trying to highlight the origin of the subject. However, when the current status of the disease/treatment/method is mentioned, only newly-published studies and reviews are accepted for a comprehensive and proper introduction.

Materials and Methods: 

This section of this review needs some major revisions and structural changes, otherwise this review cannot be accepted as a systematic review.

Authors need to carefully read the author’s instructions and adjust their manuscript accordingly. The author’s instructions of Gels, states that the “Materials and Methods” section must be placed after “Results and Discussion” and “conclusion” sections. 

Any proper systematic review must respect the standard requirements and structural properties listed below:

1. Systematic reviews are highly suggested to be registered at PROSPERO.

2. Following a well-established universal guideline (e.g., PRISMA 2020, and Cochrane) is a must. Otherwise, that study simply would not be a systematic review and would be more of a literature review. 

3. The study question of the review according to the PICO format, the PROSPERO registration ID, and a declaration for following the PRISMA guidelines must all be included in the “methods” section of the review. 

4. All of the following titles and subtitles must be included in the “methods” section of a proper systematic review: eligibility criteria, types of studies, types of participants/study groups, types of interventions, control group, primary and secondary outcomes, information sources, search strategy, study selection, inclusion and exclusion criteria, data collection, quality assessments, and synthesis methods.

5. Preparing a PRISMA flow diagram for the screening and inclusion process of studies, is highly suggested for all systematic reviews. 

If a review does not fulfill all of the mentioned requirements, then it simply would not be a “systematic review”. I strongly advice authors reconstruct their whole manuscript if they persist to present/submit it as a systematic review.

Results:

Similar to the “methods” section, authors have failed to comply to any of the key guidelines for a proper “results” section of a systematic review. 

Discussion:

Since authors’ methodology and the style of reporting the results are not appropriate for a systematic review, their results and their judgements, simply cannot be reviewed/judged. Readers and reviewers can only judge a systematic review as long as it has done a decent job in designing a search strategy, inclusion/exclusion criteria and overall methodology. This review has failed to comply to any of the mentioned requirements. 

Overall, I believe authors have done a poor job in executing a proper systematic review. No guidelines are obeyed/applied in this study. This study has not been registered at PROSPERO. I suggest authors reconstruct/rewrite their whole manuscript from the start  designing a proper search strategy and inclusion/exclusion criteria  in order to have a proper systematic review. Another option would be authors presenting their study as a review of literature, and give up on the whole idea of a systematic review. I suggest rejecting this review with no revisions allowed/accepted.  

Author Response

REPLY to Reviewer 3

Authors have conducted this review on the current status of the assessments of MSCs and biocompatible scaffolds for bone and tissue regenerations. They have mentioned in both the abstract and the introduction that this is a systematic review. However, as mentioned below – section by section –, I believe that this review has simply not been conducted in a “systematic” manner.

 REPLY: We thank the Reviewer for the observation. Considering overall the suggestions regarding the type of manuscript, we propose to consider this work a Review, deleting in the text the word “Systematic”.

Abstract:

Lines 27, 28, and 29:

“Data were extracted covering background trial information including location, clinical phase, condition, follow-up, control used, MSCs tissue source and characterization, and scaffolds used.”:

I find including your primary and secondary outcomes in the abstract unnecessary. These details belong to the “methods” and the “results” sections.

REPLY: We thank the Reviewer for the suggestion. The sentence has been modified as follow: “Data were extracted covering background trial information” (LINE 27-28)

Instead, I highly suggest authors detail their study questions according to the PICO format in their abstract so that authors and reviewers would have a proper comprehension of the reviewers’ main goals and concerns.

REPLY: We thank the Reviewer for the observation. We followed the IMRAD structure as reported in MDPI guidelines. However, following also the above suggestions of the Reviewer, the abstract has been modified (line 23-40).

Lines 29, 30, 31, and 32:

“Overall, these findings highlight the importance and the efficacy of cell-scaffolds constructs in bone regenerative medicine in different conditions. The majority of studies used calcium phosphate-based scaffolds and hydrogels for tissue regeneration, while bone marrow was the primary source of cells.”

I strongly advice not using vague and undetailed sentences and reports in your abstract. If authors want to report any part of their results in the abstract, the proper way would be including actual numbers or percentages instead of descriptive vague statements.

REPLY: We agree with the Reviewer. We detailed the number of studies that used scaffolds and source of cells in the abstract. The following sentence has been added in the Abstract: “ The majority of scaffolds are calcium phosphate ceramic alone, such as b-tricalcium phosphate (TCP) (2 clinical trials), biphasic calcium phosphate bioceramic granules (3 clinical trials), anorganic bovine bone (2 clinical trials), while bone marrow was the primary source of MSCs (5 clinical trials).” (line 33-36)

Keywords:

I suggest adding “hydrogels” to the list of keywords.

REPLY: We agree with the Reviewer. We added “hydrogels” in the keywords. (line 43)

Introduction:

  • Lines 40 and 41:

“Musculoskeletal health problems due to osteoporosis, tumors, and fractures are widely studied in recent years and bone is the most frequently transplanted tissue”. Please refer to a newly published study (preferably a review of literature) for this statement.

REPLY: Following the suggestion of the Reviewer, references number 1,2 have been added:

1- Manzini BM, Machado LMR, Noritomi PY, et al. Advances in Bone tissue engineering: A fundamental review. J Biosci 2021;46(17). PMID: 33737501

2- Mitchell SAT, Majuta LA, Mantyh PW. New Insights in Understanding and Treating Bone Fracture Pain. Curr Osteoporos Rep 2018;16(4):325-332, doi:10.1007/s11914-018-0446-8

  • Lines 41 and 42:

“Each year an estimated 2.2 million individuals suffer a fracture due to bone disease”. Since this statement directly indicated the epidemiologic state of the mentioned disease/accident, I highly suggest authors include more studies that are newer than reference “1”. Authors do not have remove reference “1”, but add newer references along with reference “1” for this epidemiologic statement.

REPLY: Following the suggestion of the Reviewer, reference number 3,4,5 have been added:

3- Starr J, Tay YKD, Shane E. Current Understanding of Epidemiology, Pathophysiology, and Management of Atypical Femur Fractures. Curr Osteoporos Rep 2018;16(4):519-529, doi:10.1007/s11914-018-0464-6

4- Hernlund E, Svedbom A, Ivergård M, et al. Osteoporosis in the European Union: medical management, epidemiology and economic burden. A report prepared in collaboration with the International Osteoporosis Foundation (IOF) and the European Federation of Pharmaceutical Industry Associations (EFPIA). Arch Osteoporos 2013;8(1):136, doi:10.1007/s11657-013-0136-1 5- Sfeir JG, Drake MT, Khosla S, et al. Skeletal Aging. Mayo Clin Proc 2022;97(6):1194-1208, doi:10.1016/j.mayocp.2022.03.011

. Lines 48, 49, 50 and 51:

“Bone regenerative medicine has been introduced into clinical practice as an alternative therapeutic approach to overcome the obstacles related to the use of current bone graft substitutes to create functional tissues instead of implanting non-living scaffolds”

The reference assessed for these statements is dated and needs to be updated.

REPLY: Following the suggestion of the Reviewer, reference number 9 has been added:

9- Di Pietro L, Palmieri V, Papi M, et al. Translating Material Science into Bone Regenerative Medicine Applications: State-of-The Art Methods and Protocols. Int J Mol Sci 2022;23(16), doi:10.3390/ijms23169493

  • Lines 74, 75 and 76:

“This approach allows the creation of an environment that drives and stimulates the cells to form new, functional tissue that subsequently integrates into the existing tissue at the defect site”

The reference assessed for these statements is dated and needs to be updated.

REPLY: Following the suggestion of the reviewer, Reference number 17 has been added:

17- Kirankumar S, Gurusamy N, Rajasingh S, et al. Modern approaches on stem cells and scaffolding technology for osteogenic differentiation and regeneration. Exp Biol Med (Maywood) 2022;247(5):433-445, doi:10.1177/15353702211052927.

  • Lines 84, 85, 86, and 87:

“The therapeutic potential of these cells is rather derived from the release of growth factors, cytokines, and extracellular vesicles to their surrounding cells that may favor bone regeneration and osteogenesis”.

The referred study for this statement is from 2012. When the therapeutic potential of MSCs is mentioned, I strongly believe that using a dated study does not do this subject any justice. Authors need to search for newly published studies on this subject and refer to them along with reference “11”.

REPLY: Thanks again for this suggestion. Reference 18 has been added:

18- Samsonraj RM, Raghunath M, Nurcombe V, et al. Concise Review: Multifaceted Characterization of Human Mesenchymal Stem Cells for Use in Regenerative Medicine. Stem Cells Transl Med 2017;6(12):2173-2185, doi:10.1002/sctm.17-0129

  • Lines 87 and 88:

“In the field of skeletal diseases, several preliminary clinical studies have demonstrated that MSCs are the most promising cell population [18,19].”

References number 18 and 19, are assessed before reference number 12. Authors need to re-adjust all of their references and their references list. I strongly advice utilizing referencing tools(e.g., Endnote) to prevent and correct these kinds of mistakes.

REPLY: We thank the Reviewer for the advice. We checked the entire references and, in particular, the numbers mentioned by the reviewer. The entire reference list has been updated.

Overall, authors need to refer to newly-published studies when the history, epidemiology, or previous works done of a particular subject are mentioned. Otherwise, their judgements and reportswould not be justifiable. In addition to the mentioned references above, references number 13, 15, 16, and 20, all need to be updated to newer studies. I do understand that sometimes referring to old studies is justifiable when authors are trying to highlight the origin of the subject. However, when the currentstatus of the disease/treatment/method is mentioned, only newly-published studies and reviews are accepted for a comprehensive and proper introduction.

REPLY: We thank the Reviewer for the suggestion. The references 12, 13, 16 and 18 are now updated with the following references:

12- Ho-Shui-Ling A, Bolander J, Rustom LE, et al. Bone regeneration strategies: Engineered scaffolds, bioactive molecules and stem cells current stage and future perspectives. Biomaterials 2018;180(143-162, doi:10.1016/j.biomaterials.2018.07.017

13- Manzini BM, Machado LMR, Noritomi PY, et al. Advances in Bone tissue engineering: A fundamental review. J Biosci 2021;46(17). PMID: 33737501. Please see lines 40,41.

16- Quan H, Ren C, He Y, et al. Application of Biomaterials in Treating Early Osteonecrosis of the Femoral Head: Research Progress and Future Perspectives. Acta Biomater 2023, doi:10.1016/j.actbio.2023.04.005

18- Samsonraj RM, Raghunath M, Nurcombe V, et al. Concise Review: Multifaceted Characterization of Human Mesenchymal Stem Cells for Use in Regenerative Medicine. Stem Cells Transl Med 2017;6(12):2173-2185, doi:10.1002/sctm.17-0129

Materials and Methods:

This section of this review needs some major revisions and structural changes, otherwise this review cannot be accepted as a systematic review.

Authors need to carefully read the author’s instructions and adjust their manuscript accordingly. The author’s instructions of Gels, states that the “Materials and Methods” section must be placed after “Results and Discussion” and “conclusion” sections.

 REPLY: Thanks for your advice. We checked and we found that in review is not mandatory the paragraph materials and methods, otherwise the article type suggest introducing this paragraph after the introduction section. We followed the IMRAD structure as reported in MDPI guidelines.

Any proper systematic review must respect the standard requirements and structural properties listed below:

  1. Systematic reviews are highly suggested to be registered at PROSPERO.
  2. Following a well-established universal guideline (e.g., PRISMA 2020, and Cochrane) is a must. Otherwise, that studysimply would not be a systematic review and would be more of a literature review.
  3. The study question of the review according to the PICO format, the PROSPERO registration ID, and a declaration for following the PRISMA guidelines must all be included in the“methods” section of the review.
  4. All of the following titles and subtitles must be included in the “methods” section of a proper systematic review: eligibility criteria, types of studies, types of participants/study groups, types of interventions, control group, primary and secondary outcomes, information sources, search strategy, study selection, inclusion and exclusion criteria, data collection, quality assessments, and synthesis methods.
  5. Preparing a PRISMA flow diagram for the screening and inclusion process of studies, is highly suggested for all systematic reviews.

If a review does not fulfill all of the mentioned requirements, then it simply would not be a “systematic review”. I strongly advice authors reconstruct their whole manuscript if they persist to present/submit it as a systematic review.

 REPLY: Considering that all the requested statements have not been fully respected as suggested by the reviewer, we choose to delete the word “systematic” and we consider this a Review.

Results:

Similar to the “methods” section, authors have failed to comply to any of the key guidelines for a proper “results” section of a systematic review.

REPLY: Considering that all the requested statements have not been fully respected as suggested by the reviewer, we choose to delete the word “systematic” and we consider this a Review.

Discussion:

Since authors’ methodology and the style of reporting the results are not appropriate for a systematic review, their results and their judgements, simply cannot be reviewed/judged. Readers and reviewers can only judge a systematic review as long as it has done a decent job in designing a search strategy, inclusion/exclusion criteria and overall methodology. This review has failed to comply to any of the mentioned requirements.

Overall, I believe authors have done a poor job in executing a proper systematic review. No guidelines are obeyed/applied in this study. This study has not been registered at PROSPERO. Isuggest authors reconstruct/rewrite their whole manuscript from the start – designing a proper search strategy and inclusion/exclusion criteria – in order to have a proper systematic review. Another option would be authors presenting their study as a review of literature, and give up on the whole idea of a systematic review. I suggest rejecting this review with no revisions allowed/accepted.

REPLY: Thank you again for this suggestion, even if the strategy design, inclusion and exclusion criteria have been described in the text, we recognized that we have not been fully respected the PRISMA guidelines, and so we choose to delete the word “systematic” and we consider this a Review of literature.

Round 2

Reviewer 2 Report

The authors took into account all the comments of the reviewer and answered the questions in detail. I believe that the Discussion section could be expanded, but this is for the consideration of the editor-in-chief. I believe that in this form the manuscript can be published in the journal Gels.

Reviewer 3 Report

Authors have conducted this review on the current status of the assessments of MSCs and biocompatible scaffolds for bone and tissue regenerations. They have mentioned in both the abstract and the introduction that this is a systematic review. However, as mentioned below  section by section , I believe that this review has simply not been conducted in a “systematic” way. 

Note: My review of their revised paper will be in the color red.

Abstract:

Lines 27, 28, and 29: 

“Data were extracted covering background trial information including location, clinical phase, condition, follow-up, control used, MSCs tissue source and characterization, and scaffolds used.”: 

I find including your primary and secondary outcomes in the abstract unnecessary. These details belong to the “methods” and the “results” sections. Instead, I highly suggest authors detail their study questions according to the PICO format in their abstract so that authors and reviewers would have a proper comprehension of the reviewers’ main goals and concerns.

Lines 29, 30, 31, and 32:

“Overall, these findings highlight the importance and the efficacy of cell-scaffolds constructs in bone regenerative medicine in different conditions. The majority of studies used calcium phosphate-based scaffolds and hydrogels for tissue regeneration, while bone marrow was the primary source of cells.”

I strongly advice not using vague and undetailed sentences and reports in your abstract. If authors want to report any part of their results in the abstract, the proper way would be including actual numbers or percentages instead of descriptive vague statements. 

Revision of the Abstract:

Authors have significantly elevated the structure of their abstract. Their reported results are less vague and more concise. The aim of the study is clearly stated. 

Keywords:

I suggest adding “hydrogels” to the list of keywords.

Revision of the Keywords:

“Hydrogel” has been added to the list of keywords as requested.

Introduction:

·      Lines 40 and 41: 

“Musculoskeletal health problems due to osteoporosis, tumors, and fractures are widely studied in recent years and bone is the most frequently transplanted tissue”. Please refer to a newly published study (preferably a review of literature) for this statement. 

Authors have updated their references for this statement. The references number 1 and 2 are relatively new and are reviews.

·      Lines 41 and 42:

“Each year an estimated 2.2 million individuals suffer a fracture due to bone disease”. Since this statement directly indicated the epidemiologic state of the mentioned disease/accident, I highly suggest authors include more studies that are newer than reference “1”. Authors do not have remove reference “1”, but add newer references along with reference “1” for this epidemiologic statement. 

Authors have upgraded their references, and now references number 3 and 5 are newly published and have been referred to properly.

·      Lines 48, 49, 50 and 51:

“Bone regenerative medicine has been introduced into clinical practice as an alternative therapeutic approach to overcome the obstacles related to the use of current bone graft substitutes to create functional tissues instead of implanting non-living scaffolds” 

The reference assessed for these statements is dated and needs to be updated.

Authors have now referred to reference number “9” and it is from 2022 which is ideal, and its content is suitable.

·      Lines 74, 75 and 76:

“This approach allows the creation of an environment that drives and stimulates the cells to form new, functional tissue that subsequently integrates into the existing tissue at the defect site” 

The reference assessed for these statements is dated and needs to be updated.

Authors have now added references number 16 and 17 which both are new and appropriate.

·      Lines 84, 85, 86, and 87:

“The therapeutic potential of these cells is rather derived from the release of growth factors, cytokines, and extracellular vesicles to their surrounding cells that may favor bone regeneration and osteogenesis”. 

The referred study for this statement is from 2012. When the therapeutic potential of MSCs is mentioned, I strongly believe that using a dated study does not do this subject any justice. Authors need to search for newly published studies on this subject and refer to them along with reference “11”.

Authors have added reference number “27” which is new and appropriately assessed.

·      Lines 87 and 88:

“In the field of skeletal diseases, several preliminary clinical studies have demonstrated that MSCs are the most promising cell population [18,19].” 

References number 18 and 19, are assessed before reference number 12. Authors need to re-adjust all of their references and their references list. I strongly advice utilizing referencing tools (e.g., Endnote) to prevent and correct these kinds of mistakes. 

The issue with the order of the references has been solved. In addition the new reference for this statement is from 2022.

Overall, authors need to refer to newly-published studies when the history, epidemiology, or previous works done of a particular subject are mentioned. Otherwise, their judgements and reports would not be justifiable. In addition to the mentioned references above, references number 13, 15, 16, and 20, all need to be updated to newer studies. I do understand that sometimes referring to old studies is justifiable when authors are trying to highlight the origin of the subject. However, when the current status of the disease/treatment/method is mentioned, only newly-published studies and reviews are accepted for a comprehensive and proper introduction.

Authors have successfully updated their references to newly-published original studies and reviews. The order of their references in the reference list has been solved. 

Materials and Methods: 

This section of this review needs some major revisions and structural changes, otherwise this review cannot be accepted as a systematic review.

Authors need to carefully read the author’s instructions and adjust their manuscript accordingly. The author’s instructions of Gels, states that the “Materials and Methods” section must be placed after “Results and Discussion” and “conclusion” sections. 

Any proper systematic review must respect the standard requirements and structural properties listed below:

1. Systematic reviews are highly suggested to be registered at PROSPERO.

2. Following a well-established universal guideline (e.g., PRISMA 2020, and Cochrane) is a must. Otherwise, that study simply would not be a systematic review and would be more of a literature review. 

3. The study question of the review according to the PICO format, the PROSPERO registration ID, and a declaration for following the PRISMA guidelines must all be included in the “methods” section of the review. 

4. All of the following titles and subtitles must be included in the “methods” section of a proper systematic review: eligibility criteria, types of studies, types of participants/study groups, types of interventions, control group, primary and secondary outcomes, information sources, search strategy, study selection, inclusion and exclusion criteria, data collection, quality assessments, and synthesis methods.

5. Preparing a PRISMA flow diagram for the screening and inclusion process of studies, is highly suggested for all systematic reviews. 

If a review does not fulfill all of the mentioned requirements, then it simply would not be a “systematic review”. I strongly advice authors reconstruct their whole manuscript if they persist to present/submit it as a systematic review.

Authors have not referred to their paper as “A systematic review” in their revised manuscript. And now they present their work as a “Concise review”, which is absolutely the right decision. Their work is simply not a systematic review and authors have understood this fact. And now their work, its structure and their judgements are more justifiable since it is not a systematic review.

Results:

Similar to the “methods” section, authors have failed to comply to any of the key guidelines for a proper “results” section of a systematic review. 

Since authors do not present their work as a systematic review anymore, then their results section can be accepted.

Overall, I believe authors have done a poor job in executing a proper systematic review. No guidelines are obeyed/applied in this study. This study has not been registered at PROSPERO. I suggest authors reconstruct/rewrite their whole manuscript in order to have a proper systematic review. Another option would be authors presenting their study as a review of literature, and give up on the whole idea of a systematic review. I suggest rejecting this review with no revisions allowed/accepted. 

Authors have understood that there are major differences between their work and a proper systematic review. They have given up on presenting their work as a systematic review, as I suggested. The current revised paper is accepted as a concise review/review of literature. I suggest accepting this review.